# Towards Labeling-free Fine-grained Animal Pose Estimation

## ABSTRACT

In this paper, we are interested in identifying denser and finer animals joints. The lack of standardized joint definitions across various APE datasets, e.g., AnimalPose with 20 joints, AP-10k with 17 joints, and TigDog with 19 joints, presents a significant challenge yet offers an opportunity to fully utilize annotation data. This paper challenges this new non-standardized annotation problem, aiming to learn fine-grained (e.g., 24 or more joints) pose estimators in datasets that lack complete annotations. To combat the unannotated joints, we propose FreeNet, comprising a base network and an adaptation network connected through a circuit feedback learning paradigm. FreeNet enhances the adaptation network's tolerance to unannotated joints via body part-aware learning, optimizing the sampling frequency of joints based on joint detection difficulty, and improves the base network's predictions for unannotated joints using feedback learning. This leverages the cognitive differences of the adaptation network between non-standardized labeled and large-scale unlabeled data. Experimental results on three non-standard datasets demonstrate the effectiveness of our method for fine-grained APE.

## CCS CONCEPTS

• **Computing methodologies → Interest point and salient region detections**; **Computer vision**; **Biometrics**; **Semi-supervised learning settings**.

## KEYWORDS

pose estimation, animal biometrics, free labeling, meta learning

## 1 INTRODUCTION

Animal pose estimation (APE) aims to localize the joint positions on animal bodies. It has important implications for a range of applications, including behavior understanding, wildlife conservation, animal individual identification, and the generation of animal-related multimedia content [2, 18, 21]. Accurate APE can also contribute to developing more immersive and interactive multimedia content involving animals, enhancing user engagement. Despite well-established techniques in human pose estimation for complex scenes [8, 26], APE is still at its infancy stage [10] due to significant appearance variance, behavior difference, and joint distribution shifts. Early studies [7, 17] attempted to train the model on a single-category APE dataset and transfer the learned knowledge to other animals. However, these methods are limited in their ability to

*ACM MM, 2024, Melbourne, Australia*

© 2024 Copyright held by the owner/author(s). Publication rights licensed to ACM.
ACM ISBN 978-x-xxxx-xxxx-x/YY/MM
https://doi.org/10.1145/nnnnnnn.nnnnnnn

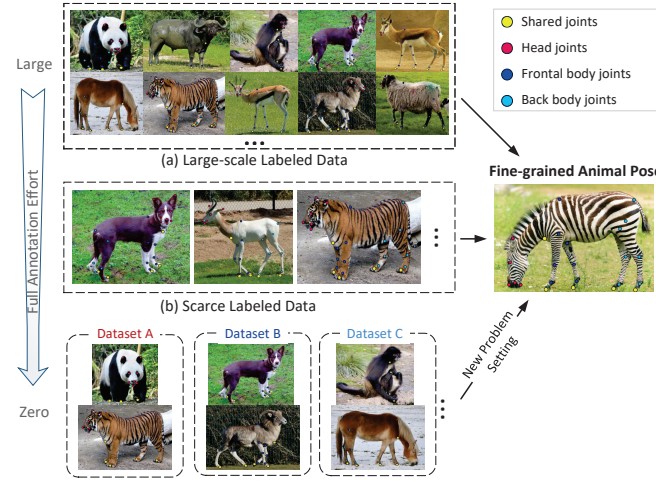

**Figure 1: Illustration of animal pose estimation task. Existing methods depend on full annotating (a) large-scale data or (b) scarce data, whereas (c) we use *zero* full annotation. We aim to learn fine-grained pose estimators from non-standard datasets, which is a new annotation problem setting in APE.**

handle species that do not share certain similarities with trained animals. Some methods [12, 19] utilize CAD models to generate synthetic animal images and labeled joints for APE. However, these models often face a significant domain gap due to the limited variation in environmental conditions and viewpoints.

Existing studies [20, 29] circumvent this domain gap by utilizing a general APE dataset with massively increased scale and animal diversity (in Figure 1(a)). For example, AnimalPose [4] contains 4,666 images, AP-10k [30] contains 10,015 images, and TigDog [6] contains 14,093 images. However, collecting large-scale datasets with precise full annotation costs a large effort (e.g., 20 seconds for one joint even using AI-assisted annotation [16]). Another feasible way is leveraging large-scale unlabeled data to alleviate the need for fully annotated labeled data. For example, ScarceNet [13] employs strategies such as reliable pseudo-label selection and reusable sample re-labeling to combat the noisy pseudo labels for APE. Although the annotation effort is getting smaller (in Figure 1(b)), full data annotation is still mandatory. *This raises a question: for achieving **fine-grained** animal pose (denser joints), do we need to re-annotate existing datasets with full annotations?* To answer this question, we focus on how to learn fine-grained animal poses from non-standard labeled data with zero full annotation as illustrated in Figure 1(c).

This non-standardized annotation problem is a *new* challenge in fine-grained APE. Nevertheless, it is significant and valuable in practical applications as there is no consensus on annotation standards across existing APE datasets. Specifically, AnimalPose [4] with 20 joints, AP-10k [30] with 17 joints, and TigDog [6] with 19 joints, presents a unique challenge and opportunity in fully utilizing

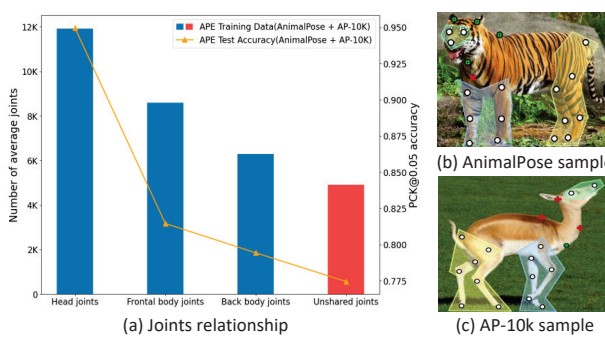

Figure 2: (a) The joint relationship across different datasets. (b-c) Non-standardized annotations in two samples, including exclusive joints in green and unannotated joints in red. We use HRNet trained with AnimalPose and AP-10k for testing.

annotation data. However, existing works rely on full annotated data and neglect to utilize multiple non-standard annotation data for APE. In this paper, we aim to address the non-standardized annotation problem. As illustrated in two samples of Figure 2, the shared joints are grouped into three body parts, namely the head, frontal body, and back body, and the exclusive joints of each dataset (marked in green) are grouped as "unshared joints". We mark the unannotated joints in red. Figure 2 (a) demonstrates the joint relationship across different datasets, which provides two key insights. First, we observe shared joints in different animal body parts exhibit different learning difficulties due to dataset gaps and considerable species variation. This accuracy difference between head joints and back body joints is close to 10%. Second, due to lacking full annotated data, the unshared joints across different datasets have the smallest average number and the worst accuracy, aggravating the skewed joint distribution in the combined training data.

Intuitively, different classifiers can produce different decision boundaries and have different learning abilities. To combat the unannotated joints, we chose bi-model training to leverage the different abilities of the two models in complementary ways. Specifically, we propose **FreeNet**, which consists of a base network and an adaptation network, aiming at fine-grained APE with **free** full annotation labels. FreeNet enhances the adaptation network's tolerance to unannotated joints via body part-aware learning and improves the base network's predictions for unannotated joints via feedback learning. Body part-aware learning is proposed to mitigate different learning difficulties of animal bodies, and the core is part-aware sampling, ensuring that easy-to-detect joints (i.e., the head joints) are sampled less and hard-to-detect joints (i.e., the back body joints) are sampled more. Our circuit feedback learning paradigm connects the base network and adaptation network and informs the base network how good the pseudo labels of unannotated joints are by using the cognitive difference of the adaptation network between non-standard labeled data (i.e., with unannotated data) and large-scale unlabeled data. As a result, the training error caused by those joints is adaptively corrected.

Two training pools, non-standard labeled and unlabeled data, are utilized to train the two networks. We refer to each fully-trained base and adaptation mode as a generation. At each generation,

1) FreeNet dynamically selects pseudo-joints from unlabeled data to facilitate body part-aware learning of the adaptation network. The joints are divided into three parts based on different animal bodies: the head, frontal body, and back body. For each part, we rank the joints based on the confidence scores produced by the base network and select the most confident joints for learning. Our body part-aware sampling determines the percentage of joints selected for each part. 2) FreeNet adopts a circuit feedback mechanism to refine the base network's predictions for unannotated joints, leveraging the cognitive differences of the adaptation network. If the cognitive difference between non-standard labeled and selected unlabeled data is significant, the base network is penalized using the adverse direction of current gradients. The selection of unlabeled data is guided by the confidence score rankings of pseudo-joints from both networks to ensure unannotated joints are further improved during feedback learning.

To sum up, this paper makes the following contributions:

- We address the non-standardized annotation problem, a *new* and significant challenge in fine-grained animal pose estimation, which is useful in real-world applications due to the lack of consensus on APE annotation standards.
- We propose FreeNet, a general framework to combat the unannotated joints via meta-optimization. FreeNet enhances the adaptation network's tolerance to unannotated joints through body part-aware learning and uses feedback learning to improve the base network's predictions of these joints.
- We achieve state-of-the-performance for fine-grained APE on non-standard datasets. Furthermore, we can learn fine-grained pose estimators without requiring full annotations.

## 2 RELATED WORK

### 2.1 Datasets for Animal Pose Estimation

Deep architectures have made remarkable progress in human pose estimation thanks to the availability of large-scale, high-quality annotated datasets like MPII [1] and COCO [15]. When it comes to animal pose estimation, early works have established datasets for specific animals such as horses [4], dogs [3], and tigers [19]. However, models trained on these datasets have poor generalization ability. To reduce the need for extensive human labor, some researchers have explored using CAD models to generate synthetic animal images for pose estimation [12, 19, 23]. However, these synthetic images have limited variations in environmental conditions and viewpoints, which leads to a significant performance gap when adapting to real-world animal images. To bridge this gap, several large-scale datasets including AP-10k [30], APT-36k [29], Animal-Pose [4], and Animal Kingdom [20] have been introduced recently. The AP-10k and APT-36k datasets were established by the same research group, with APT-36k focusing on video-based animal pose tracking. The Animal Kingdom dataset provides multiple annotated tasks to facilitate understanding animal behavior. However, there is no consensus on the annotation standards, and the definition of animal joints differs across datasets, as shown in Table 1. For instance, the AP-10k dataset includes 54 animal species with 17 joints, while the AnimalPose dataset covers only 5 species but with 20 joints. The lack of standardization in defining animal poses brings a challenge yet opportunity in fully exploiting annotation data.

**Table 1: Analysis of various animal pose datasets.**

|  | TigDog | AnimalPose | AP-10k |
|---|---|---|---|
| Species | 2 | 5 | 54 |
| Image | 14,903 | 4,666 | 10,015 |
| Instance | NA | 6,117 | 13,028 |
| Joint | 19 | 20 | 17 |
| Exclusive joints:9 | neck | ✗ | neck |
|  | ✗ | nose | nose |
|  | ✗ | left ear | ✗ |
|  | ✗ | right ear | ✗ |
|  | ✗ | throat | ✗ |
|  | ✗ | wither | ✗ |
|  | chin | ✗ | ✗ |
|  | shoulder-like(L) | ✗ | ✗ |
|  | shoulder-like(R) | ✗ | ✗ |
| Shared joints:15 | left eye, right eye, root of tail, left shoulder, left elbow, left frontal paw, right shoulder, right elbow, right frontal paw, left hip, left knee, left back paw, right hip, right knee, right back paw | | |

## 2.2 Animal Pose Estimation

Pose estimation can be viewed as a regression of heatmaps, and ideas such as bottom-up and top-down approaches for human pose estimation can also be adapted for animals [10]. However, due to the significant differences between humans and animals, the development of APE methods heavily relies on the availability of APE datasets. When only specific animal pose data is available, the focus is on transferring specific domain knowledge to a more generalized animal context [12, 19, 23]. However, a considerable gap still exists, especially when dealing with unseen animal species [14]. To address cross-species generation for APE, D-Gen [14] enhances animal pose estimation by breaking the inconsistent relations among joints while preserving the consistent ones. Synthetic animal images generated from CAD are then used to facilitate APE by reducing the domain gap. CC-SSL [19] employs three consistency criteria to constrain both spatial and temporal to general pseudo labels. To correct noisy pseudo labels, MDAM [12] gradually updates the pseudo labels to prevent network overfitting. However, synthesizing realistic images is limited due to environmental and viewpoint variations. To reduce human labor, ScarceNet [13] learns APE with scarce but full annotations. In contrast, Our FreeNet focuses on learning from several non-standard datasets to achieve fine-grained animal pose estimation with zero full annotation data.

## 2.3 Learning with Unlabeled Data

Annotating animal images can be labor-intensive, error-prone, and hard to maintain semantic consistency for joints. To address this problem, semi-supervised learning with unlabeled data can be feasible. General image classification methods such as PL [11] and noisy student [27] generate pseudo labels for unlabeled data from model predictions. UDA [24] improves semi-supervised learning by incorporating data augmentation [5] to limit the invariance of model predictions to input noise. FixMatch [24] simplifies the learning process by training the model with high-confidence pseudo labels.

MPL [22] enables the teacher network to adjust based on students' performance feedback on labeled data, which improves pseudo labels. GPGML [28] proposes inexact supervised meta-learning to use coarse-grained labels of training samples to reduce the need for labeled data. To mitigate the negative effect of unannotated joints while facilitating the learning of shared joints, we combine PL with body part-aware sampling and feedback learning.

## 3 OUR METHOD: FREENET

### 3.1 Overview

*3.1.1 Preliminaries.* Let's consider our training set $\mathcal{D}$, comprising unlabeled data $\mathcal{D}_u = \{(x_u)\}$ and $M$ non-standard datasets with labeled images, denoted as $\mathcal{D}_l = \{\mathcal{D}_{l_1} \cup \mathcal{D}_{l_2} ... \cup \mathcal{D}_{l_M}\}$. The labeled data $\mathcal{D}_l$ forms a joint sample space $\mathcal{X}_l \times \mathcal{Y}_l^*$, where $\mathcal{X}_l$ is the image space and $\mathcal{Y}_l^*$ is the corresponding label space with partial annotations (i.e., some joints are missing annotations). Specifically, the $i$-th labeled dataset $\mathcal{D}_{l_i} = \left\{\left(x_{l_i}, y_{l_i}^*\right)\right\}$ comprises $N_i$ instance samples. Here, $N_{\mathcal{J}_i}$ denotes the total number of joints for all samples, and $\mathcal{J}_i$ refers to the definition of joints. By combining these $M$ non-standard datasets, we obtain animal poses characterized by a richer set of semantic joints, denoted by $\mathcal{J}_l = \{\mathcal{J}_1 \cup \mathcal{J}_2 ... \cup \mathcal{J}_M\}$, where $\mathcal{J}_i^s$ represents the shared joints, and $\mathcal{J}_i^e$ denotes the exclusive joints unique to the labeled dataset, e.g., chin for TigDog and wither for AnimalPose. Therefore, the unannotated joints for $\mathcal{D}_{l_i}$ can be represented by $\mathcal{J}_l \setminus (\mathcal{J}_i^s \cup \mathcal{J}_i^e)$. We propose the FreeNet framework, which leverages several non-standard datasets and unlabeled data to address the unannotated joints.

*3.1.2 Our framework.* FreeNet consists of an adaptation network ($\mathcal{A}$) and a base network ($\mathcal{B}$), both utilizing the same network architecture (e.g., HRNet [25]) but with independent weights. This bi-model training helps combat unannotated joints by leveraging the different abilities of the two models. We learn FreeNet by training the adaptation network and the base network sequentially in each generation. The training objective for the adaptation network is denoted by

$$\mathcal{L}_{\mathcal{A}} = \mathcal{L}_u,$$

where $\mathcal{L}_u$ guides to learn body part-aware features from pseudo-labels. Meanwhile, the overall training objective for the base network is:

$$\mathcal{L}_{\mathcal{B}} = \mathcal{L}_s + \mathcal{L}_f,$$

where $\mathcal{L}_s$ supervises learning of prior features from non-standard datasets, and the feedback loss $\mathcal{L}_f$ refines the base network through feedback learning. Figure 3 illustrates the FreeNet pipeline, with non-standard labeled and selected unlabeled data serve as inputs on the left. We avoid merging non-standard and pseudo labels in a single training pool to prevent poor performance and convergence failure due to their label distribution disparity (see Figure 4).

At each generation, the base network generates pseudo labels for the unlabeled data $\hat{\mathcal{D}}_u = \{(x_u, \hat{y}_u)\}$. The adaptation network then uses a selected pseudo-labeled subset $\hat{\mathbb{U}} \subset \hat{\mathcal{D}}_u$ (achieved by applying body part-aware sampling) for training. The base network learns prior knowledge from several non-standard labeled data. With feedback learning, the base network uses another selected pseudo-labeled subset $\hat{\mathbb{S}}_{feedback} \subset \hat{\mathcal{D}}_u$ (i.e., using a threshold $\alpha_{feedback}$)

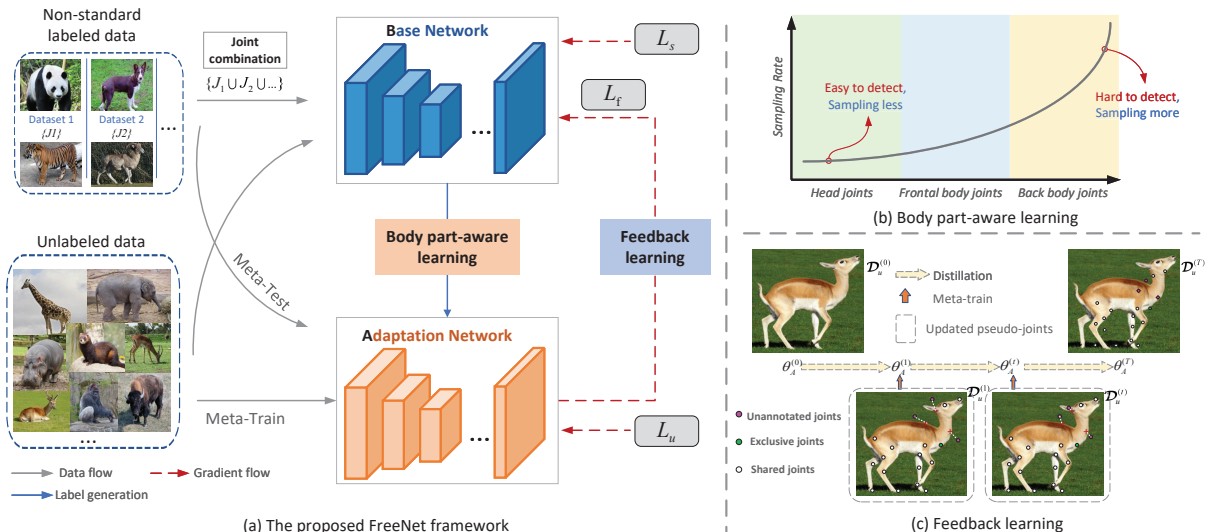

Figure 3: (a) The framework of FreeNet, trained on non-standard labeled and selected unlabeled data. In the meta-train phase, FreeNet enhances the adaptation network's tolerance to unannotated joints via (b) body part-aware learning. In the meta-test phase, FreeNet improves the base network's ability to accurately predict unannotated joints via (c) feedback learning.

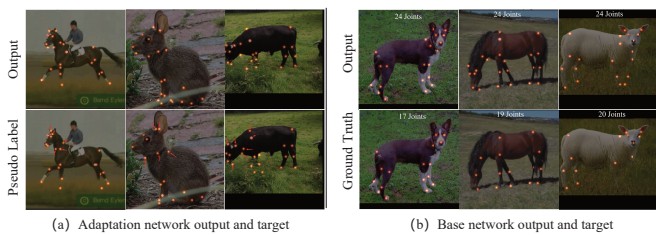

Figure 4: We utilize (a) pseudo labels and (b) non-standard labels as training targets. There are substantial difference in their label distribution. The arrow highlights the generated unannotated joints.

to generate better pseudo-joints, especially for unannotated joints, therefore improving the performance of the adaptation network. Ultimately, the adaptation network outperforms the base network in the circuit feedback paradigm. During the inference stage, only the adaptation network is used to estimate animal poses with denser joints. The following parts will detail the different learning modules and loss computation.

## 3.2 Body Part-aware Learning

Joints in different animal bodies often exhibit varying learning difficulties and can adversely hurt performance if used directly for training. To address this, FreeNet dynamically selects pseudo-joints from unlabeled data to facilitate body part-aware learning in the adaptation network. We first decompose the joints into three animal body parts: the head, frontal body, and back body. Based on their confidence scores determined by the base network, we rank the joints for each part and select the most reliable for learning. We intentionally avoid using joints with small losses for training, as

relying on a "small loss" criterion [9] can lead to a skewed distribution of joints, which hinders learning a good adaptation network. Our body part-aware sampling strategy determines the sampling frequency of joints based on joint detection difficulty. It prioritizes hard-to-detect joints by sampling these more frequently than those easy-to-detect joints, as shown in Figure 3(b).

The proportion of joints to be selected for each body part is determined by thresholds $\gamma_{head}$, $\gamma_{frontal}$, and $\gamma_{back}$. For instance, a head joint will be selected if its corresponding confidence score falls within the $\gamma_{head}$ proportion of all sorted confidence scores for head joints. The selected pseudo-joints corresponding to the head, frontal body and back body are included in $\hat{\mathbb{U}}$ to update the adaptation network using $\mathcal{L}_u$:

$$\mathcal{L}_u = \frac{\sum_{k=1}^{N_J} \left\{ \hat{H}_u^k \in \hat{\mathbb{U}} \right\}_1 \mathcal{L}_u^k}{\sum_{k=1}^{N_J} \left\{ \hat{H}_u^k \in \hat{\mathbb{U}} \right\}_1},$$

$$\text{where } \mathcal{L}_u^k = \left\| \hat{H}_u^k - (\mathcal{A}(x_u; \theta_{\mathcal{A}}))^k \right\|^2. \quad (1)$$

$\hat{H}_u^k$ represents the pseudo heatmap for the $k$-th joint, which can be derived from soft prediction $\mathcal{B}(x_u; \theta_{\mathcal{B}})$ in two steps: 1) extracting joints from soft prediction with highest confidence; 2) applying 2D Gaussian centered on each joint location with a standard deviation of 1 pixel. $\mathcal{L}_u^k$ is the corresponding pseudo label based loss for the $k$-th joint. $\{\text{condition}\}_1$ is a condition function, which outputs 1 when the condition is true and 0 otherwise.

## 3.3 Non-standard Datasets Learning

The base network is trained with non-standard labeled datasets. We apply a joint combination to transform partially annotated labeled spaces, $\mathcal{Y}_l^*$, into fully annotated ones, $\tilde{\mathcal{Y}}_l$. In this transformation, we use $\mathbf{0}$ to represent joints that lack annotations, resulting in a comprehensive labeled dataset $\tilde{\mathcal{D}}_l$. Specifically, for each joint set $J_i$,

we define $J_i^{'} = (J_i, \mathbf{0})$, where $\mathbf{0}$ signifies the unannotated joints. The dimension of $\mathbf{0}$ corresponds to the number of unannotated joints, denoted as $\| J_i^{'} \setminus (J_i^s \cup J_i^e) \|$.

We use ground truth heatmaps, $H_l$, derived from $\tilde{\mathcal{Y}}_l$, as targets for training the base network. The training process aims to minimize the mean square error (MSE) between the predicted heatmaps and these target heatmaps, which can be expressed as:

$$\mathcal{L}_s = \sum_{i=1}^{M} \left\| H_{l_i} - \mathcal{B}(x_{l_i}; \theta_{\mathcal{B}}) \right\|^2 \tag{2}$$

where $M$ is the number of non-standard datasets and $M \geq 2$.

## 3.4 Feedback Learning

To combat the unannotated joints, we propose feedback learning that aims to improve the base network's ability to predict pseudo-joints, especially those that are unannotated. The intuition behind the base network update is the relationship between the "new" adaptation network on non-standard labeled data and the "old" adaptation network on unlabeled data. Essentially, the adaptation network gauges the cognitive differences between labeled and unlabeled data to update the base network as feedback. If the gradients of two networks have the same direction, the base network is updated in the current direction. In contrast, if the gradients of two networks have different directions, the base network is punished using the adverse direction of current gradients. As the base network is improved to produce accurate predictions for unannotated joints, this strategy can help narrow the gap between non-standard labeled and large-scale unlabeled data. Using more accurate pseudo labels will further improve the performance of the adaptation network for APE.

The base network is trained on non-standard labeled data that includes unannotated joints, whereas the adaptation network is trained using unlabeled data with fully pseudo-labels. In view of this, we use the confidence score rankings of pseudo-joints from both networks as the selection criterion. The threshold $\alpha_{feedback} = (a\%, b\%)$ determines the proportion of pseudo-joints selected for $\hat{\mathbb{S}}_{feedback}$. The criteria are as follows,

(1) Pseudo-joints ranked between $a\%$ and $b\%$ by either the base network or the adaptation network are included in $\hat{\mathbb{S}}_{feedback}$.
(2) Pseudo-joints with conflicting ranking between the two networks are also selected in $\hat{\mathbb{S}}_{feedback}$. Specifically, we focus on those with ranking falling smaller than $a\%$ or greater than $b\%$ in either network.

This selection process is rational because pseudo-joints confirmed as high-confidence by two networks do not require further updates. Conversely, pseudo-joints identified as low-confidence by both networks are excluded from feedback learning to avoid performance degradation. Once $\hat{\mathbb{S}}_{feedback}$ is established, the feedback loss is calculated as the dot product of two terms:

$$\mathcal{L}_f = f \cdot \frac{\sum_{k=1}^{N_J} \left\{ \hat{H}_u^k \in \hat{\mathbb{S}}_{feedback} \right\}_1 \mathcal{L}_f^k}{\sum_{k=1}^{N_J} \left\{ \hat{H}_u^k \in \hat{\mathbb{S}}_{feedback} \right\}_1}, \tag{3}$$

$$\text{where } \mathcal{L}_f^k = \left\| \hat{H}_u^k - (\mathcal{B}(I_u; \theta_{\mathcal{B}}))^k \right\|^2.$$

The first term $f$ is the feedback coefficient that determines the direction and strength of the update; the second term is the loss

of the base network on selected unlabeled data $\hat{S}_{feedback}$. $\mathcal{L}_f^k$ is the corresponding loss for the $k$-th joint. Specifically, the feedback coefficient $f$ is defined as:

$$f = \eta_{\mathcal{A}} \cdot (\nabla_{\theta_{\mathcal{A}}^{(t+1)}} \text{MSE}(H_l, \mathcal{A}(x_l; \theta_{\mathcal{A}}^{(t+1)}))^{\top} \cdot$$
$$\nabla_{\theta_{\mathcal{A}}} \text{MSE}(\hat{H}_u, \mathcal{A}(x_u; \theta_{\mathcal{A}}^{(t)}))), \tag{4}$$

where $f$ is calculated as a dot product of two terms: the gradients of the "new" adaptation network on non-standard labeled data and the gradients of the "old" adaptation network on large-scale unlabeled data. The sign of $f$ will determine the direction of the update, while the absolute value of $f$ will determine its strength. The adaptation network uses selected pseudo-joints data to update the parameters to $\mathcal{A}^{(t+1)}$. In particular, we approximate it with the parameters obtained from $\mathcal{A}^{(t)}$ by updating the base network parameters on $(x_u, \hat{H}_u)$, i.e., $\theta_{\mathcal{A}}^{(t+1)} = \theta_{\mathcal{A}}^{(t)} - \eta_{\mathcal{A}} \nabla_{\theta_{\mathcal{A}}} \text{MSE}(\hat{H}_u, \mathcal{A}(x_u; \theta_{\mathcal{A}}))$.

## 3.5 Algorithm for FreeNet

We listed detailed step-by-step pseudo-code for FreeNet in Algorithm 1. FreeNet learns fine-grained pose estimators by extracting rich APE knowledge from non-standard labeled and large-scale unlabeled data. This is achieved by body part-aware learning and a circuit feedback paradigm. Each generation of FreeNet involves the following steps: 1) Select unlabeled images from $\hat{\mathbb{U}}$ based on whether the pseudo-joint confidence score ranking of a specific animal body part meets the sampling thresholds: $(\gamma_{head}, \gamma_{frontal}, \gamma_{back})$. 2) The adaptation network is first updated in line 10 by minimizing the unsupervised loss $\mathcal{L}_u$, on selected unlabeled data $\hat{\mathbb{U}}$. This facilitates knowledge transfer from the base network $\mathcal{B}$ to the adaptation network $\mathcal{A}$ through the generation of pseudo ground truths, conditioned on body part-aware sampling. 3) The base network is then updated in line 22 with two losses: the supervised loss $\mathcal{L}_s$ (line 13) and the feedback loss $\mathcal{L}_f$ (line 18). The two losses guide the learning process of the base network, as illustrated in line 15 and line 20, respectively. Specifically, an unlabeled image is selected from $\hat{\mathbb{S}}_{feedback}$ to trigger the feedback loss $\mathcal{L}_f$ when a pseudo-joint is moderately ranked based on the confidence score of entire joints in the batch. This selection process improves the base network's predictions for unannotated joints. By this design, FreeNet enables the base network and adaptation network to continuously enhance and complement each other.

## 4 EXPERIMENTS

### 4.1 Datasets and Implementation Details

*4.1.1 Datasets.* Our experiments are conducted on three widely used benchmark datasets for fine-grained APE. **AP-10k** [30] is a large-scale benchmark that contains 10,015 labeled images across 54 species, each annotated with 17 joints. These images are divided into train, validation, and test sets with a ratio of 7:1:2 for each species. **AnimalPose** [4] contains 4,666 labeled images from 5 animal species: cat, dog, sheep, cow, and horse, annotated with 20 joints. **TigDog** includes 6,523 tiger images and 8,380 horse images, each annotated with 19 joints. We provide a detailed joint definition for each dataset in Table 1.

**Algorithm 1** Training Procedure of FreeNet

---

**Input**: Non-standard labeled data $\mathcal{D}_l = \left\{ \left( \mathcal{X}_l, \mathcal{Y}_l^* \right) \right\}$ and unlabeled data $\mathcal{D}_u$;
Body part-aware sampling thresholds: $\gamma_{head}, \gamma_{frontal}, \gamma_{back}$;
Feedback learning threshold: $\alpha_{feedback}$;
**Outputs**: $\Theta_{\mathcal{A}}^{(T)}$
**Initialize**: $\theta_{\mathcal{B}}^{(0)}$ and $\theta_{\mathcal{A}}^{(0)}$

1: Apply joint combination: $\tilde{\mathcal{Y}}_l \leftarrow \mathcal{Y}_l^*$
2: Get combined labeled data: $\tilde{\mathcal{D}}_l \leftarrow \mathcal{D}_l$
3: **for** $t = 0...T - 1$ **do**
4:     $x_l, H_l \leftarrow$ SampleMiniBatch($\tilde{\mathcal{D}}_l$)
5:     $x_u \leftarrow$ SampleMiniBatch($\mathcal{D}_u$)
6:     $\hat{H}_u \leftarrow$ Forward($x_u, \theta_{\mathcal{B}}^{(t)}$)
7:     Obtain $\hat{\mathbb{U}}$ by using thresholds ($\gamma_{head}, \gamma_{frontal}, \gamma_{back}$)
8:     **if** $x_u \in \hat{\mathbb{U}}$ **then**
9:         Compute the loss $\mathcal{L}_u$ according to Eqn. (1)
10:        Update the adaptation network by pseudo label
11:        $\theta_{\mathcal{A}}^{(t+1)} \leftarrow \theta_{\mathcal{A}}^{(t)} - \eta_{\mathcal{A}} \nabla_{\theta_{\mathcal{A}}} \text{MSE}(\hat{H}_u, \mathcal{A}(x_u; \theta_{\mathcal{A}}))$
12:     **end if**
13:     Compute the loss $\mathcal{L}_s$ according to Eqn. (2)
14:     Compute the base network's gradient on combined labeled data
15:     $g_{\mathcal{B},s}^{(t)} \leftarrow \nabla_{\theta_{\mathcal{B}}} \text{MSE}(H_l, \mathcal{B}(x_l; \theta_{\mathcal{B}}))$
16:     Obtain $\hat{\mathbb{S}}_{feedback}$ by using threshold $\alpha_{feedback}$
17:     **if** $x_u \in \hat{\mathbb{S}}_{feedback}$ **then**
18:        Compute the loss $\mathcal{L}_f$ according to Eqn. (3-4)
19:        Compute the base network's gradient via feedback
20:        $g_{\mathcal{B},f}^{(t)} \leftarrow f \cdot \nabla_{\theta_{\mathcal{B}}} \text{MSE}(\hat{H}_u, \mathcal{B}(x_u; \theta_{\mathcal{B}}))$
21:     **end if**
22:     Update the base network:
23:     $\theta_{\mathcal{B}}^{(t+1)} \leftarrow \theta_{\mathcal{B}}^{(t)} - \eta_{\mathcal{B}} \cdot (g_{\mathcal{B},s}^{(t)} + g_{\mathcal{B},f}^{(t)})$
24: **end for**
25: **return** $\Theta_{\mathcal{A}}^{(T)}$

---

*4.1.2 Evaluation metrics.* We adopt the Object Keypoint Similarity (OKS) as the evaluation metric and represent the mean Average Precision (mAP) across OKS=0.50,0.55,...0.90,0.95 following [25]. Additionally, we include the Percentage of Correct Keypoints (PCK) metric [19], which quantifies the percentage of joints accurately predicted within a normalized distance from the ground truths.

*4.1.3 Experimental protocols.* To evaluate the precision of additional learned joints in fine-grained animal poses, we consider two settings: 1) using synthetic datasets derived from AP-10k and 2) employing a combination of real datasets. The synthetic datasets are generated by segmenting AP-10k into subsets, each containing an equal number of images but varying joint definitions to mimic the non-standard annotation problem in real-world scenarios. During testing, we utilize AP-10k's ground-truth joints to independently evaluate the accuracy of the unannotated and shared joints, as well as their accuracy gap. In the case of the real dataset combination, we report results for 10% of the two combined datasets (AP-10k and

**Table 2: Comparing with SOTA methods on scarce and non-standard datasets. ips means images per species.**

| Settings | Full Annot. | Methods | mAP↑ | PCK@0.05↑ |
|---|---|---|---|---|
| 5 ips | ✓ | ScarceNet | 53.3 | 65.2 |
| 25 ips | ✓ | ScarceNet | 68.1 | 78.2 |
| | ✗ | ScarceNet | 55.04 | 66.26 |
| 3 synthetic | ✗ | UDA | 50.8 | 64.06 |
| datasets | ✗ | FixMatch | 43.8 | 57.56 |
| from 25 ips | ✗ | MPL | 50.7 | 63.51 |
| | ✗ | Ours | **57.9** | **68.31** |

AnimalPose) and 10% of the three combined datasets (AP-10k, AnimalPose, and TigDog) to demonstrate the efficacy of our FreeNet in terms of body part-aware learning and feedback learning. The combination of real datasets is used as the experimental protocol in the ablation study.

*4.1.4 Training details.* All images are augmented using random scaling, rotation, horizontal flip, and a half-body mask and then resized to 256×256 pixels. More details are provided in the supplementary material. The size of output heatmaps is 64×64 pixels. We use HRNet-w32, trained on non-standard labeled data for 210 epochs, as the default backbone for both the base and adaptation networks. The learning rates are first initialized (i.e., 1e-5 for the base network and 1e-3 for the adaptation network) and further decayed with a cosine annealing strategy. Our batch size is 64, comprising 8 labeled and 56 unlabeled samples. Unless otherwise stated, the default thresholds ($\gamma_{head}, \gamma_{frontal}, \gamma_{back}$) for body part-aware sampling are set at (0.9, 0.75, 0.6). The default feedback learning threshold $\alpha_{feedback}$ is (20%, 80%). We set different training steps for different settings to fully train the model: 30,000 for AP-10k, 60,000 for the 10% combination of two datasets, and 60,000 for the 10% combination of three datasets. It is trained end-to-end with two NVIDIA V100 GPUs.

## 4.2 Comparison with State-of-the-Art Methods

We compare our method with four state-of-the-art semi-supervised learning methods, including UDA [24], FixMatch [24], MPL [22] and ScarceNet [13]. We re-implemented UDA, FixMatch, and MPL using open-source repositories originally developed for classification tasks. These methods are selected as they align well with our aim of leveraging unlabeled data to improve performance when labeled data is limited. Additionally, we include ScarceNet, a SOTA method for APE with scarce annotations, using its available source code for training and evaluation.

Given the recent success of ScarceNet in APE with limited data, we evaluate our method in the ScarceNet setting on the synthetic dataset of AP-10k. The $\gamma_{head}, \gamma_{frontal}$, and $\gamma_{back}$ thresholds for the body part-aware sampling are 0.5, 0.6, and 0.65, respectively. The feedback learning threshold $\alpha_{feedback}$ is (20%, 80%). Table 2 shows the comparison results with state-of-art methods. We focus on estimating animal poses from datasets with scarce, non-standardized annotations, specifically under the "25 images per species" setting (including 1250 images and 1710 instances), where ScarceNet

**Table 3: Results comparison on synthetic datasets derived from AP-10k when varying the size of shared joints.**

| Methods | Shared Joint Size | | |
|---|---|---|---|
| | 5 | 9 | 17 (full annot.) |
| mAP/PCK@0.05 on entire 17 joints | | | |
| SL | 71.39/79.87 | 72.71/81.39 | 74.47/82.72 |
| Ours | **72.70/80.57** | **74.20/82.50** | NA |
| PCK@0.05 and accuracy gap on shared/unannotated joints | | | |
| SL | 87.90/77.50 | 86.24/77.78 | 82.72/NA |
| | 10.4 | 8.46 | NA |
| Ours | **87.90/79.20** | **87.00/79.50** | NA |
| | **8.7** | **7.5** | NA |

achieves the best accuracy. However, this setting remains unfair to other comparison methods and our approach, since we do not assume scarce labels in the problem set, and these images are tailored to favor the performance of ScarceNet. Our three synthesized non-standard datasets contain the head, frontal body, and back body as exclusive joints, respectively, with "neck" and "root of tail" as shared joints. The total number of joints is equals to that in the "5 images per species" setting used by ScarceNet. All compared methods have the same inference time of 0.023 seconds per image as they have the same backbone model with 28.5M parameters. In Table 2, The results present two findings: 1) Non-standard datasets pose more significant challenges than scarce data with complete annotations, as most methods underperform the "5 images per species" setting. 2) Our method outperforms all other methods, demonstrating the effectiveness of FreeNet in handling both scarce and non-standardized annotations.

## 4.3 Evaluation on non-standardized annotation

To verify the effectiveness of our FreeNet for non-standardized annotations, we report mAP and PCK@0.05 accuracy on two synthetic datasets from AP-10k in Table 3. Specifically, we use the accuracy as a function of the size of the shared joints, ranging from 5, 9 to 17. The two synthetic datasets, AP-10k-subA and AP-10k-subB, have different exclusive joints. AP-10k-subA has exclusive joints from the frontal body of animals, while AP-10k-subB has exclusive joints from the back body of animals. 1) "5" shared joints include the left/right eye, nose, neck, and root of the tail, which coarsely reflect the animal's length. 2) "9" shared joints expand this by adding the left/right frontal paw and left/right back paw, which coarsely constrains the bounding box of the animal. 3) "17" shared joints represent the full annotation of AP-10k for supervised learning, setting an upper bound performance. The results demonstrate that FreeNet consistently outperforms SL in mAP and PCK@0.05 metrics. Moreover, FreeNet achieves more balanced results between shared and unannotated joints, leading to a smaller accuracy gap. Specifically, in the "5" shared joint setting, FreeNet shows a smaller accuracy gap of 8.7 compared to the SL method's 10.4. Similarly, in the "9" shared joint setting, FreeNet maintains a smaller accuracy gap of 7.5 compared to the SL method's 8.46.

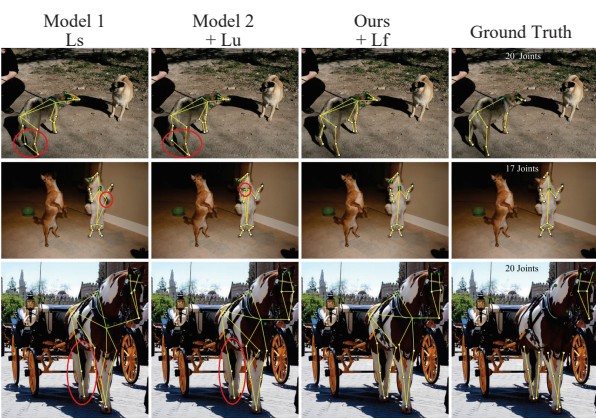

**Figure 5: Qualitative examples for FreeNet with different loss functions. Compared to the left column, the baseline model in the right column adds a new loss by using '+'.**

**Table 4: Performance evaluation of FreeNet on 10% combined AP-10k and AnimalPose using different loss functions.**

| Models | Loss | | | mAP↑ | PCK@0.05↑ |
|---|---|---|---|---|---|
| | $\mathcal{L}_s$ | $\mathcal{L}_u$ | $\mathcal{L}_f$ | | |
| 1 | ✓ | ✗ | ✗ | 52.2 | 67.6 |
| 2 | ✓ | ✓ | ✗ | 56.2 | 70.63 |
| Ours | ✓ | ✓ | ✓ | **57.26** | **71.36** |

**Table 5: Performance of the body part-aware learning with different sampling ratios.**

| $\gamma_{head}$ | $\gamma_{frontal}$ | $\gamma_{back}$ | mAP↑ | PCK@0.05↑ | std PCK↓ |
|---|---|---|---|---|---|
| 100% | 100% | 100% | 56.9 | **71.4** | 6.6 |
| 90% | 75% | 60% | 56.2 | 70.63 | 7.8 |
| 50% | 60% | 65% | **57.5** | 71.28 | **6** |

## 4.4 Ablation Study

*4.4.1 Effect of FreeNet design.* Table 4 presents the performance of FreeNet using different losses on combined real datasets, which includes 10% of AP-10k and AnimalPose. The results indicate that (1) FreeNet achieves the best performance using three losses together, compared to the first two models, demonstrating its effectiveness in addressing the non-standardized annotation problem. (2) The unsupervised loss $\mathcal{L}_u$ enhances animal pose estimation by a large margin, verifying the effectiveness of body part-aware learning. (3) Incorporating the feedback learning loss $\mathcal{L}_f$ updates the base network to generate better heatmaps on pseudo-joints, especially for unannotated joints, further improving the results. Figure 5 provides some quantitative examples of predicted samples generated by different baselines, which align with the quantitative results.

*4.4.2 Effect of body part-aware learning.* As illustrated in Table 5, using all pseudo-joints of unlabeled data, i.e., $\gamma_{head} = \gamma_{frontal} = \gamma_{back} = 100\%$, does not lead to optimal accuracy as different animal

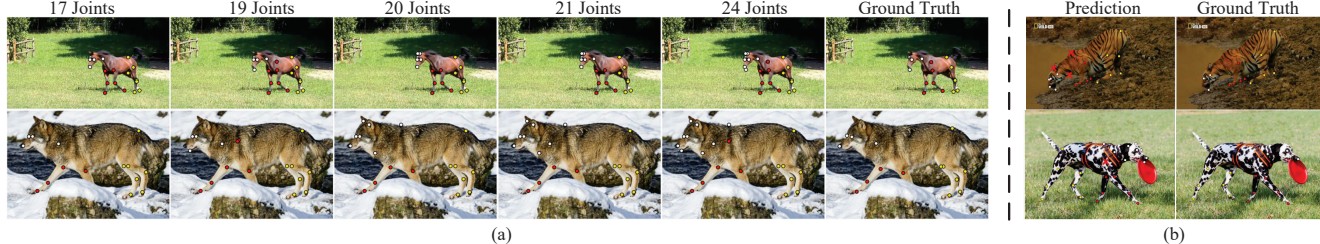

Figure 6: (a) FreeNet's potential for generating denser joints. (b) Failure cases, red crosses mark undetected unannotated joints.

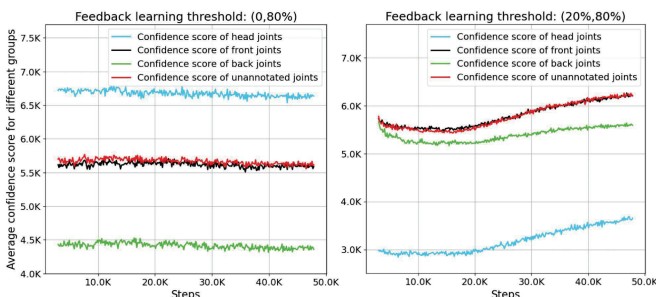

Figure 7: Feedback learning with (20%,80%) proportion largely improves the prediction confidence for unannotated joints.

Table 6: Performance of our feedback learning with different thresholds to address unannotated joints estimation.

| $\alpha_{feedback}$ | (0,100%) | (0,80%) | (20%,80%) |
|---|---|---|---|
| mAP↑ | 56.47 | 56.30 | **57.26** |
| PCK@0.05↑ | 70.60 | 70.80 | **71.36** |

body parts exhibit different learning difficulties. Moreover, using the traditional "small loss" criterion in our method is also inappropriate as it tends to exaggerate the imbalanced number of joints, resulting in decreased mAP and PCK@0.05 accuracy. For instance, settings of $\gamma_{head} = 90\%$, $\gamma_{frontal} = 75\%$, and $\gamma_{back} = 60\%$ underperform compared to the setting using all pseudo-joints. In contrast, our body part-aware learning module effectively balances the learning difficulty of joints in different parts. Table 5 shows that we achieve more balanced accuracy (6% std in PCK@0.05) across different animal body parts and also yield higher APE accuracy by sampling 50% of head joints, 60% of frontal body joints, and 65% of back body joints. It's worth noting that further tuning these empirical sampling thresholds could lead to even higher accuracy.

*4.4.3 Insights of feedback learning.* Proper learning of unannotated joints is essential to improve fine-grained APE. Table 6 shows the impact of using different proportions of pseudo-joints in feedback learning. Notably, employing pseudo-joints with confidence scores ranking from 20% to 80% yields the highest accuracy. Another interesting finding is that excluding joints with the lowest confidence scores does not affect the performance, as (0,100%) and (0,80%) thresholds produce similar results. However, discarding

joints with the highest confidence scores, from (0,80%) to (20%,80%), can enhance network learning and increase performance. This is because these excluded joints are already well-detected and do not need further feedback learning, as illustrated in Figure 7(a). In contrast, our feedback learning at (20%,80%) significantly improves the prediction confidence for unannotated joints by training more steps as illustrated in Figure 7(b). These results are consistent with the quantitative results in Table 6 and demonstrate our feedback learning under (20%,80%) can better learn unannotated joints.

## 4.5 Qualitative Results on Real-World Applications

We use our FreeNet to identity denser and finer joints in real-world applications where there is no consensus on APE annotation standards. Specifically, we expanded two non-standard datasets by adding a 10% TigDog dataset for training. As illustrated in Figure 6(a), FreeNet increases the number of joints from 17, 19, and 20 to 21, 24 without manual annotating. This is particularly useful, as it saves up to 140 seconds per animal by eliminating the need for additional joint labeling. By adding more non-standard datasets, FreeNet has the potential to identify more than 24 animal joints. We believe that FreeNet provides a general framework that can leverage arbitrary nonstandard annotation data for other pose estimation related tasks (e.g., human/hand pose estimation) where consistent high-quality data are lacking.

## 5 LIMITATIONS & CONCLUSION

Our approach achieves impressive performance in identifying denser joints even when only a few non-standard annotations are available. However, there are still several challenges remaining for fine-grained APE. The network sometimes has difficulty generalizing well to some unannotated joints for certain species (e.g., tigers) that only appear in one dataset, since animal species vary greatly across datasets (see Figure 6(b)). We leave this for further research.

In conclusion, this paper tackles the emerging and significant challenge of non-standardized annotations in APE. Our solution, FreeNet, aims to fully utilize these annotations from multiple non-standard datasets to learn denser joints at no additional labeling cost. Specifically, we address the dilemma of unannotated joints and facilitate the learning of joints shared across datasets. Experimental results on synthetic and combined real datasets demonstrate the effectiveness of FreeNet, which is useful in real-world applications.

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
