# OpenReview forum: "Towards Labeling-free Fine-grained Animal Pose Estimation"
_acmmm.org/ACMMM/2024/Conference — MM2024 Poster_

### Official Review · Reviewer_Zt6j · 2024-05-15

**Rating:** 3
**Confidence:** 2

**Summary:**

This work presents a new framework called FreeNet that addresses the challenges of animal pose estimation (APE) using datasets with non-standardized annotations and without requiring full annotation. FreeNet utilizes a base network and an adaptation network, enhancing tolerance to unannotated joints through body part-aware learning and improving prediction accuracy with feedback learning. This approach is tested across multiple non-standard datasets, demonstrating its ability to efficiently learn fine-grained animal poses.

**Strengths:**

- FreeNet introduces a unique strategy combining body part-aware learning and feedback learning to handle unannotated and non-standardized data effectively.
-  By leveraging datasets with partial annotations and enhancing learning through its dual-network architecture, FreeNet reduces reliance on fully annotated datasets.
-  The method has been shown to improve the accuracy of pose estimation in non-standard datasets by addressing the annotation inconsistencies and exploiting the available data more comprehensively.

**Limitations:**

- The starting point is good and the authors are trying to solve a new problem, which is interesting. I coarsely understand the problem setting of this work. However, the statements about the specific method are not clear. For example, I do not understand the training process or inference process according to Figure 3. It is the best way for the authors could clearly state the specific steps such as i) ii) iii) balabala...
-  Another question is about the feedback learning. How do you update theta given data D in T steps? Is such an update simple to execute? The authors also show the distillation process in Figure 3 (c), which confused me as to why such a process is a distillation process.
-  The paper could also benefit from a more detailed discussion on scenarios where FreeNet underperforms, including potential limitations in handling extremely diverse or rare species not represented in the training data.
-  Figure 6 is not clear to recognize which one is better. The authors could enlarge the image size or highlight the differences.
-  Table 2 shows comparisons with SOTA methods. It is better to show the model complexity for a fair comparison.

**Suitability:**

2

---

### Official Review · Reviewer_niDm · 2024-05-24

**Rating:** 5
**Confidence:** 2

**Summary:**

In this paper, the authors address the challenge of non-standardized joint definitions in various Animal Pose Estimation (APE) datasets. They propose FreeNet, a method that aims to learn fine-grained pose estimators (with 24 or more joints) in datasets lacking complete annotations.

**Strengths:**

- This paper addresses the challenge of non-standardized annotation in fine-grained animal pose estimation, crucial for real-world applications due to the lack of consensus on APE annotation standards.
- This paper introduce FreeNet, a framework that enhances the adaptation network's tolerance to unannotated joints and improves the base network's predictions through meta-optimization and feedback learning.

**Limitations:**

- Will different categories of keypoints cause a long-tail effect?
- Define the missing keypoints as 0, how is the gradient for this part propagated?
- The network can predict denser joints, but do they all possess the properties of keypoints? For example, the top-most and bottom-most keypoints on the wolf's neck in Figure 6(a) – what do these points represent?

**Suitability:**

2

---

### Official Review · Reviewer_UPbE · 2024-05-24

**Rating:** 3
**Confidence:** 4

**Summary:**

The main idea of this paper is to address the challenge of non-standardized annotations in fine-grained animal pose estimation (APE). The authors propose a novel framework called FreeNet to tackle the problem of estimating denser and finer animal joints, especially when complete annotations are lacking.

**Strengths:**

1. Non-Standardized Annotation Problem:
   - The paper identifies and tackles the challenge of non-standardized joint definitions across APE datasets, which is significant for real-world applications where consistent annotations are lacking.
2. FreeNet Framework:
   - Structure: Composed of a base network and an adaptation network linked through a circuit feedback learning paradigm.
   - Body Part-Aware Learning:This technique optimizes sampling frequency based on joint detection difficulty, improving the adaptation network’s tolerance to unannotated joints.
   - Feedback Learning: Enhances the base network's predictions for unannotated joints by leveraging cognitive differences between labeled and unlabeled data.
3. Meta-Optimization for Unannotated Joints:
   - FreeNet utilizes meta-optimization to select pseudo-joints from unlabeled data dynamically.
   - The method ensures effective learning by sampling joints based on their detection difficulty and confidence scores.

**Limitations:**

1.	The paper lacks fluency in expression and clarity in logic, which need further improvement.
2.	The paper contains few formulae, and the explanations of the existing ones are insufficient.
3.	The paper frequently mentions pseudo-joints and pseudo-label, but their specific forms are not clearly defined. Could you provide further clarification?
4.	In Figure 3, the framework for FreeNet is overly simplistic. It would be beneficial to provide a more detailed diagram of the specific network.
5.	There are issues with the experimental design. In section 4.1.3, what are the selection criteria for the non-standard labeled data? Are there targeted experiments or risks of overfitting? What would be the effects of using different datasets or combinations?

**Suitability:**

2

---

### Meta-Review · Area_Chair_kMek · 2024-06-27

**Recommendation:** Accept (Poster)
**Confidence:** 5

**Metareview:**

This paper initially received a mix of reviews. However, after the rebuttal, reviewers upgraded their review and agreed to accept the paper. The AC thoroughly considered the paper, reviews, rebuttal, and subsequent discussions, ultimately deciding to accept it. The authors are encouraged to incorporate the reviewers' suggestions to enhance the final version.